# Radon Implicit Field Transform (RIFT): Learning Scenes from Radar Signals

## Abstract

Data acquisition in array signal processing (ASP) is costly because achieving high angular and range resolutions necessitates large antenna apertures and wide frequency bandwidths, respectively. The data requirements for ASP problems grow multiplicatively with the number of viewpoints and frequencies, significantly increasing the burden of data collection, even for simulation. Implicit Neural Representations (INRs) — neural network-based models of 3D objects and scenes — offer compact and continuous representations with minimal radar data. They can interpolate to unseen viewpoints and potentially address the sampling cost in ASP problems. In this work, we select Synthetic Aperture Radar (SAR) as a case from ASP and propose the ***Radon Implicit Field Transform*** (RIFT). RIFT consists of two components: a classical forward model for radar (Generalized Radon Transform, GRT), and an INR based scene representation learned from radar signals. This method can be extended to other ASP problems by replacing the GRT with appropriate algorithms corresponding to different data modalities. In our experiments, we first synthesize radar data using the GRT. We then train the INR model on this synthetic data by minimizing the reconstruction error of the radar signal. After training, we render the scene using the trained INR and evaluate our scene representation against the ground truth scene. Due to the lack of existing benchmarks, we introduce two main new error metrics: ***phase*-Root Mean Square Error** (p-RMSE) for radar signal interpolation, and ***magnitude-Structural Similarity Index Measure*** (m-SSIM) for scene reconstruction. These metrics adapt traditional error measures to account for the complex nature of radar signals. Compared to traditional scene models in radar signal processing, with only 10% data footprint, our RIFT model achieves up to 188% improvement in scene reconstruction. Using the same amount of data, RIFT is up to $3\times$ better at reconstruction and shows a 10% improvement generalizing to unseen viewpoints.

## 1 Introduction

Array signal processing (ASP) is a subdomain of digital signal processing (DSP) which involves multiple spatially distributed sensors (Swindlehurst et al., 2014). For some ASP problems – in particular, for imaging and detection problems – the cost of data acquisition is often high because of the relationship between resolution requirements, aperture size, and signal bandwidth. The angular resolution achieved by an array is a direct consequence of the aperture size, and the range resolution depends on the total bandwidth of the received signal (Richards, 2022; Moccia & Renga, 2011; Liu et al., 2021). The data acquisition cost grows linearly with each the number of samples, the aperture size (assuming antennas are Nyquist-spaced), and the number of frequency bins. In this work, we use radar imaging as a representative of ASP problems and address the cost of data acquisition with a deep learning model. Such combinations of machine learning and radar signal processing have been used in autonomous vehicles (Bilik et al., 2019), robotics (Ali et al., 2014), and geographic information systems (Javali et al., 2021).

Synthetic Aperture Radar (SAR) is a specialized radar imaging technique that synthesizes a large virtual antenna aperture by moving the sensor relative to the scene (Moreira et al., 2013). This process involves the coherent processing of successive radar echoes received at multiple points along the sensor path to reconstruct a high-resolution image. Most often, the radar samples are uniformly spaced throughout the synthetic aperture, so the size of the aperture determines the amount of view-

point samples radar takes. In order to give a concrete example of the amount of data needed in SAR imaging, according to NASA, for a satellite with C-band radar, to get a spatial resolution of 10 m, the synthetic radar aperture size needs to be of the size of 47 soccer fields ($\sim$ 5km) (NASA, 2023). The resulting amount of data can be on the order of terabytes.

A potential remedy to the cost of data acquisition for SAR imaging is learning based reconstruction of objects and scenes. One example is Implicit Neural Representations (INR), which involves a neural network learning scene properties[1] (colors, opacity, and so on) through measurement signals like pictures. The prospect of INR interpolating between different view points, particularly for visual data, is accomplished with different scene representation mechanisms, e.g., voxels (Choy et al., 2016), point clouds (Achlioptas et al., 2018), meshes (Kanazawa et al., 2018) and especially the occupancy network by Mescheder et al. (2019). More recently, Neural Radiance Fields (NeRF) by Mildenhall et al. (2020) integrates a physical process called light field rendering from Levoy & Hanrahan (1996) to improve model performance. NeRF sparked works illustrate that the integration of underlying physical mechanisms enables better learning and scene representations.

In this study, we integrate deep learning methods with a traditional forward model for radar signals called the Generalized Radon Transform (GRT) (Nolan & Cheney, 2002; Monga et al., 2018). Analogous to the light marching used in NeRF, the GRT is the physical mechanism integrated in the rendering process for radar, so the model can learn scene reconstruction directly from the observed radar signals. We denote our architecture as ***R**adon **I**mplicit **F**ield **T**ransform* (RIFT).

The main contributions of this work are as follows:

- We present the first method to learn implicit scene representations directly from radar signals.
- Using our method, we achieve better scene reconstruction and viewpoint interpolation with fewer measurements than traditional algorithms.
- We formulate the first joint benchmark for both radar scene reconstruction and signal interpolation which aligns with perceived quality.

## 2 BACKGROUND

### 2.1 ARRAY SIGNAL PROCESSING AND (SYNTHETIC APERTURE) RADAR

Array signal processing (ASP) generally refers to the use of two or more antennas for coherent processing. ASP is a fundamental technique and has diverse applications in radar, sonar, and communications. The use of multiple transmitters or receivers enhances overall system performance by increasing gain, enabling beamforming, providing spatial filtering, and increasing signal-to-noise ratio (SNR) (Van Veen & Buckley, 1988). As wireless spectrum becomes more crowded and systems evolve to utilize higher frequency bands (Berger, 2014) – e.g., in 5G networks – the use of larger and more sophisticated antenna arrays has become crucial for achieving the precise beamforming necessary for efficient communication.

The basic principle of ASP in the narrowband setting involves adjusting the phase and amplitude of signals received by (or transmitted from) each element in the array. For signals that originate far from the antenna array, the spherical wavefront impinging on the antenna array appears locally as a plane wave. Coherently summing signals from any given direction can be accomplished by weighting the signal received at each element and adding up the signals over the array. The weights are simple phase shifts which depend on the array geometry (e.g., linear, planar, circular, etc.) and the direction-of-arrival (DoA) of the incoming signal. The phase adjustment allows for constructive interference in desired directions and destructive interference in others, effectively shaping the radiation pattern of the array. DoA estimation considers the signal angle as an unknown and tries to find the angle which best explains an observed signal.

Synthetic aperture radar (SAR) is related to DoA estimation in that goal is to sense an unknown environment. There are two important modifications, however. First, DoA estimation assumes that the signals exist in space and are being passively observed by the array. On the other hand, SAR

---

[1]In this work, the scene property of interest is complex reflectivity of the radar signal of the scene.

techniques use a transmit antenna to excite the scene and observe reflections. A second modification that distinguishes SAR from conventional radar is the motion of the antenna array relative to the scene. As the array is moving, pulses are repeatedly transmitted, and reflections are stored at a variety of viewpoints. SAR processing takes these measurements and uses array position information to form a large "synthetic" aperture. Even with a small antenna array, the path traced by the antenna can be orders of magnitude larger, leading to greatly improved imaging capabilities (Moreira et al., 2013).

## 2.2 GENERALIZED RADON TRANSFORM (GRT)

GRT is a widely-used forward model for a radar signal. It is a simplification of the Maxwell's equations under the Born approximation and planar wave assumption (Nolan & Cheney, 2002). Using Born's approximation, we discretize the scene to voxel reflectors with position $\boldsymbol{x} \in \mathbb{R}^3$. For each voxel reflector, there is an associated complex-value reflectivity $\rho(\boldsymbol{x}) \in \mathbb{C}$. $\rho(\boldsymbol{x})$ is discretized to a look-up table for the INR to learn and interpolate.

Let $s_{\text{TX}}$ and $s_{\text{RX}}$ represent the slow-time variables corresponding to the positions of the TX and RX, respectively. Let $\gamma(s)$ denote the trajectory of the antennas, and let $R_b(\mathbf{x})$ be the range function under the bistatic configuration. Consider the fast-time temporal frequency $\omega$ within the range $[\omega_{\text{Lo}}, \omega_{\text{Hi}}]$, where $\omega_{\text{Lo}}$ and $\omega_{\text{Hi}}$ are the lowest and highest frequencies used by our radar system, respectively. Each frequency $\omega$ corresponds to a wave number $k(\omega)$ according to the standard definition. We define the GRT operator $\mathcal{F}$ such that the perceived radar signal $d(\omega, s)$ can be expressed as:

$$d(\omega, s) := \mathcal{F}[\rho] \approx \int_{\mathbb{R}^3} e^{j(k(\omega)R_b(\boldsymbol{x}))} \boldsymbol{A}(\omega, s, \boldsymbol{x}) \rho(\boldsymbol{x}) e^{j\Phi(\rho(\boldsymbol{x}))} d\boldsymbol{x} \tag{1}$$

here the range function

$$R_b(\boldsymbol{x}) := ||\boldsymbol{x} - \gamma(s_{TX})|| + ||\boldsymbol{x} - \gamma(s_{RX})|| \tag{2}$$

and the function $\Phi(\rho(\boldsymbol{x}))$ stands for the phase of $\rho(\boldsymbol{x})$. The phase of reflector $\Phi(\rho(\boldsymbol{x}))$ corresponds to possible phase change takes place when the electromagnetic wave interacts with the scene. The matrix $\boldsymbol{A}$ here corresponds to the amplitude of the transmitted electromagnetic wave by the radar and it is up to a global normalization among all radar signals, as for the detail and proof, please refer to Section 3.1 and Appendix B.3, respectively.

## 2.3 IMPLICIT NEURAL REPRESENTATION

INR is a class of methods that learn a scene or an object through parameterized signals, providing a continuous interpolation that maps the signal to its domain. Compared to traditional grid-based representations, INR is more compact, as the spatial resolution in grid-based methods is inherently tied to the grid's granularity.

Following the integration of light marching by NeRF (Mildenhall et al., 2020), there is a resurgence of research into INR trained on visual data to demonstrate concrete improvement of speed in training (Garbin et al., 2021), accuracy (Barron et al., 2021), and generalizability across viewpoints (Barron et al., 2022). The above works demonstrated data efficiency, training speed and reconstruction accuracy and set cornerstones to our work.

## 3 METHODS AND EXPERIMENTAL SETUP

In this section, we present the relevant details of the GRT, the design of RIFT, and the error metrics we customize for radar data modality. The INR from RIFT takes as input the location in the scene and returns the complex reflectivity of the scene at that point. Upon receiving the reflectivity estimate from the INR, the GRT directly produces the radar signal at different viewpoints. The predicted signal is compared to the signal corresponding to the ground truth scene (referred to as "*true signal*" in this work), and automatic differentiation with gradient descent accumulates gradients to update the estimation of the scene.

### 3.1 GENERALIZED RADON TRANSFORM (GRT) AND RADAR SIGNAL SYNTHESIS

Without loss of generality, in this research, we normalize all the magnitude of radar signal $\mathbf{S}$ to $[0, 1]$. To accurately simulate real-world scenarios, we use the bistatic radar configuration mentioned in Section 2.2 In the case, the transmitter (TX) and receiver (RX) are spatially separated at each time step. Further details of the radar setup are provided in Appendix B.1.

The combination of the INR learning $\rho(\boldsymbol{x})$ and the GRT generating the radar signal constitutes the inverse problem relative to the forward signal generation. We define the *granularity* as the distance between neighboring voxels along the coordinate axes. The *granularity* is only a parameter for scene discretization. Unless otherwise noted, the scene extends a 3D cubical space with edges of 10m. The granularity of the forward problem to 0.2m, and that of the inverse problem to 0.4m. This twofold difference in granularity between the forward and inverse problems is make the RIFT more compact.

It is important to note that the dependence of the matrix $\boldsymbol{A}$ on $\boldsymbol{x}$ makes our assumption of no loss of generality nontrivial. To address this $\boldsymbol{x}$ dependence, we consider two distinct cases: the near-field approximation and the far-field approximation. However, since this pertains to an ASP problem and an extensive discussion would be tangential to the primary objective of developing a neural radar reconstruction algorithm, we defer the justification of the normalization and the difference in approaches for near and far-field approximations to Appendix B.3.

### 3.2 IMPLICIT NEURAL REPRESENTATION AND THE RIFT WORKFLOW

In this study, we assume that the scene $\rho(\mathbf{x})$ remains constant over time. The INR we employ is a function $\hat{\rho}_{\Theta}(\mathbf{x}) : \mathbb{R}^3 \to \mathbb{C}$, which parameterizes the scene's properties—specifically, the complex reflectivity for radar signals—using tunable parameters $\Theta$. Consequently, we can formulate the optimization problem as follows[2]:

$$\arg \min_{\Theta} ||\mathbf{S} - \mathcal{F}[\hat{\rho}_{\Theta}(\boldsymbol{x})]||_2 \tag{3}$$

Note that given the radar setup in Appendix B.1, the radar signal $\mathbf{S}$ is fixed. The formulation of our radar signals are in Appendix B.2. The approximation $\hat{\rho}$ utilized in this study is based on a multi-layer perceptron (MLP) model (Bishop, 1995), configured in two distinct ways: one incorporating layer normalization (Ba et al., 2016) and the other employing positional encoding, which is discussed in detail later. These configurations are designated as RIFT(N) and RIFT(S), respectively. Both models are trained using standard backpropagation techniques (Rumelhart et al., 1986), with the exact architecture and training parameters outlined in Appendix B.4.

The positional encoding configuration for INR was first introduced in NeRF (Mildenhall et al., 2020) and subsequently analyzed by Tancik et al. (2020). In this study, we adopt a mathematically equivalent structure known as SIREN (Sitzmann et al., 2020) for positional encoding within the INR framework.

In Figure 1, we present a workflow chart for RIFT. The RIFT workflow models the physical process of radar sensing, where transmitted and received waves interact with the scene. The scene is discretized into a look-up table, serving as the ground truth for our Implicit Neural Representation (INR) to learn from. RIFT comprises two main components: a GRT Segment and an INR scene model.

The hyperparameter tuning is a significant component to the training of RIFT. The details about hyperparameter tuning is elaborated in Appendix B.7.

### 3.3 ERROR METRICS AND BENCHMARK

To our best knowledge, there are no existing benchmarks to gauge how well a neural net (NN) learns both the radar signal and the corresponding scene properties. In traditional SAR imaging algorithms, common error metrics include norm-based measures like Mean Square Error (MSE) (Gonzales & Woods, 2008), structural measures like the Structural Similarity Index Measure (SSIM) (Wang et al.,

---

[2]In our experiments, we slightly modify the optimization process to enhance numerical convergence properties. Details are provided in Appendix B.5.

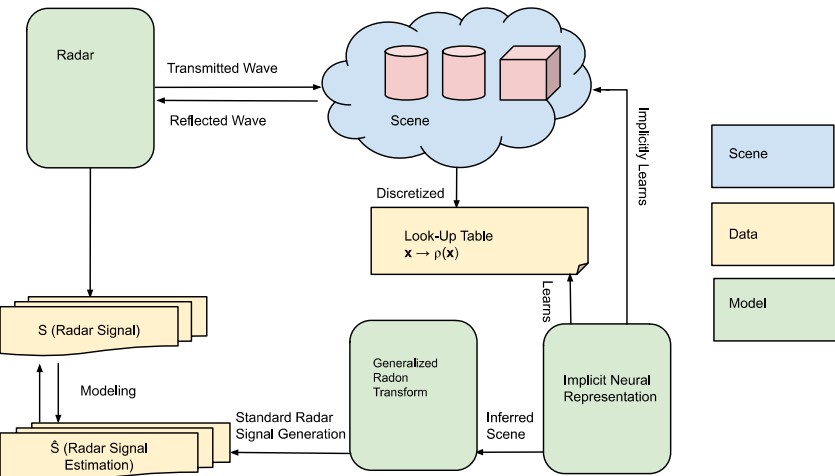

Figure 1: *Workflow chart of the RIFT architecture. The diagram illustrates how RIFT models physical radar sensing, transforms the learned scene into radar signals through the GRT segment, and iteratively refines the scene representation via backpropagation.*

2004), and probabilistic measures mainly used for classification tasks, such as Kullback-Leibler (KL) divergence (Gao, 2010).

However, although we used Mean Absolute Error (MAE) during the training stage of the INR (for an empirically better numerical stability in our experiments, please refer to Appendix B.5 for details), we cannot always distinguish between good and bad reconstructions using MAE alone. This is because the coherent addition of phase information of the signal determines the geometry, as discussed in (Franceschetti & Lanari, 1999), but the scene is determined by the magnitude of reflectivity.

Therefore, we introduce two modified traditional error metrics: magnitude-SSIM (m-SSIM), magnitude-cosine similarity (m-COS), threshold Intersection-over-Union (tIoU), and phase-Root Mean Square Error (p-RMSE). The idea is splitting the metrics to two parts: magnitude and phase. The magnitude-based metrics are used for scene reconstruction, and phase-based metrics are used for radar signal interpolation for unseen viewpoints. The reason for inducing the p-RMSE is to make up for significant variance in the magnitude of radar signal causing the model to learn only a small fraction of the training data. The detailed definition and justifications are deferred to the Appendix B.6

As a benchmark for scene rendering, we use the inverse GRT operator $\mathcal{F}^{-1}$, since there is no existing machine learning algorithm that renders a reflective scene from radar signals. We select a block version of the Kaczmarz method (Kaczmarz, 1993) as our inversion technique because it allows us to compute the inversion using a least squares formulation with a reasonably fast convergence rate. Both RIFT and the Kaczmarz method utilize radar signals without downsampling in the transmitter/receiver (TX/RX) combinations or frequency bandwidth. Due to the differing granularity between the forward signal synthesis and the inverse scene reconstruction and unseen viewpoint interpolation, we resize the scene in the forward problem to match the granularity of the scene in the inverse problem using the scikit-image library (van der Walt et al., 2014) before calculating the benchmark.

## 4   EXPERIMENTS, RESULTS AND DISCUSSION

The evaluation of RIFT's performance is twofold, addressing the core challenge of reducing data acquisition costs in SAR systems. RIFT offers two primary solutions. The first is that by reconstructing the scene using fewer radar measurements, RIFT lowers the demand for extensive data collection. This is particularly beneficial in scenarios where data acquisition is time-consuming or resource-intensive. The effectiveness of scene reconstruction is evaluated using error metrics detailed in Section 3.3, which compare the ground truth scene reflectivity with the reconstructed

scene. Then, RIFT can interpolate radar signals between previously unseen viewpoints, effectively increasing the available data supply without additional measurements. This capability enhances the system's ability to generate comprehensive radar maps from limited viewpoints. The performance of unseen viewpoint interpolation is assessed by comparing the GRT of the learned scene with the *true signal* in the test set.

## 4.1 SCENE RECONSTRUCTION

For simplicity, in this section we assume that the materials of the scenes are perfect reflectors whose phase interactions are captured by $\mathcal{F}$; that is, $\Phi(\rho(\boldsymbol{x})) = 0$. We generate three simple objects and two complex scenes, and present our results and evaluation of scene reconstruction and unseen viewpoint interpolation. All data are generated from 51 uniformly sampled azimuth and elevation angles from $[0, 2\pi]$ and $[0, \pi]$, respectively, of the spherical coordinate system described in Appendix B.1. In total, there are 2,601 possible viewpoints in the dataset we generate. When presenting results in this section, the "viewpoints" are sampled from these 2,601 viewpoints.

We demonstrate that in all cases, RIFT models use at most 40% of the training data used by the baseline model, yet they perform better in scene reconstruction by at least 109.6% in the m-SSIM metric. In all but one case mentioned in Appendix A, RIFT models outperform the baseline model in unseen viewpoint interpolation, thus achieving better generalization with significantly less data in terms of p-RMSE. Empirically, the RIFT(N) models without SIREN-style positional encoding perform better in scene reconstruction, while the RIFT(S) models perform well in cases where RIFT(N) models converge to local minima and excel at viewpoint interpolation.

### 4.1.1 SIMPLE SCENE RECONSTRUCTION

The three simple objects are a sphere of radius $2m^3$, a cube with an edge length of 2m, and a tetrahedron (denoted as "Pyramid") with a base measuring 2m by 2m and a height of 2m. Both the sphere and the cube are centered at the origin of the coordinate system described in Appendix B.1.

In Table 1[4], we compare the m-SSIM score and p-RMSE value, along with auxiliary metrics such as t-IoU and the cosine similarity of the reconstructed magnitude (denoted as m-COS). We reconstructed the scene using 100 and 1,000 viewpoints with the RIFT workflow; the corresponding datasets are denoted as RIFT(N or S)-(100 or 1000). Additionally, we applied the least squares reconstruction using 100, 200, 500, and 1,000 viewpoints; these datasets are denoted as LS-100, LS-200, LS-500, and LS-1000, respectively.

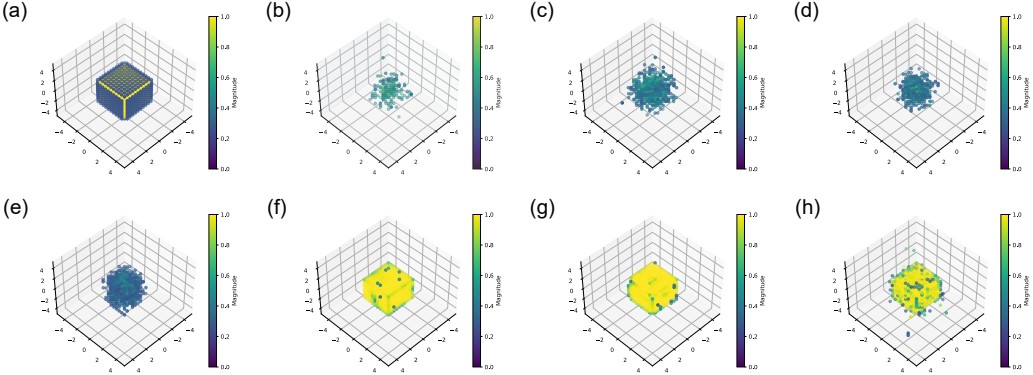

Figure 2: *Visualizations of the "cube" scene (a): Ground truth of a cube of edge of 2m. (b)-(e): Scene reconstruction by the baseline with 100, 200, 500, and 1000 viewpoints, respectively. (f): Scene reconstruction by RIFT(N) with 100 The m-SSIM score and p-RMSE of reconstruction in (f) is 0.6395 and 5.4986, 274% and 11% better than those of reconstruction in (e) while only using 10% of the viewpoints, respectively. (g), (h) Scene reconstruction by RIFT(N/S) with 1000 viewpoints as references.*

---

[3]The length units are all meters in this work unless noted otherwise.

[4]In this work, we use bold font to highlight the best performances.

Table 1: Simple Scene Reconstruction Result for Cube corresponding to Figure 2

| Model | m-SSIM | m-COS | t-IoU | p-RMSE |
|---|---|---|---|---|
| LS-100 | 0.0704 | 0.5698 | 0.0578 | 0.0155 |
| LS-200 | 0.0908 | 0.6689 | 0.1663 | 0.0153 |
| LS-500 | 0.1128 | 0.7822 | 0.1243 | 0.0151 |
| LS-1000 | 0.1706 | 0.8511 | 0.2183 | 0.0150 |
| RIFT(N)-100 | **0.6395** | 0.9792 | 0.3677 | 0.0147 |
| RIFT(S)-100 | 0.1435 | 0.6957 | 0.3302 | 0.0152 |
| RIFT(N)-1000 | 0.6298 | **0.9833** | **0.3714** | **0.0145** |
| RIFT(S)-1000 | 0.6045 | 0.9606 | 0.3688 | 0.0147 |

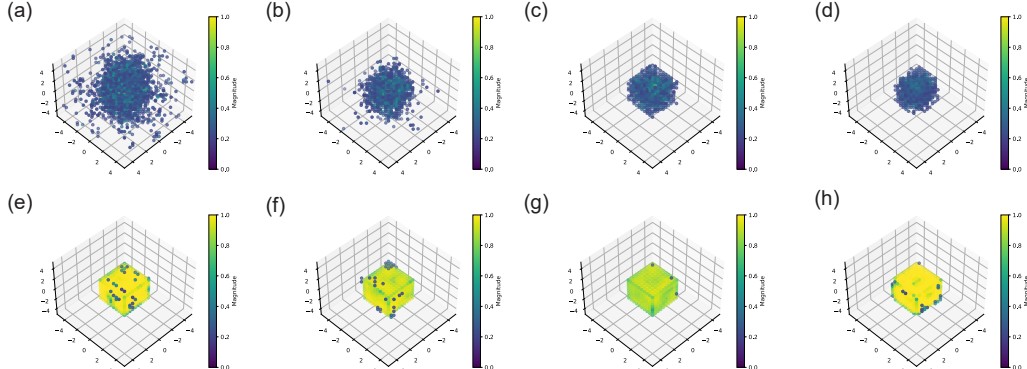

Figure 3: *Visualizations for presenting the need of data from different models. (a)-(d) Scene reconstruction by the baseline least square model with 100, 200, 500, and 1000 viewpoints. (e)-(h) Scene reconstruction by our RIFT(N) model with 100, 200, 500, and 1000 viewpoints.*

In addition to comparing the RIFT models using small amounts of data against baseline models using more data, we present Figure 3, which visualizes a comparison of two reconstructions with the same number of inputs. To ensure a fair comparison, the assumed SNR for the visualization of all eight sub-graphs is set to 0.2. The data footprint of the baseline model demonstrates the necessity for more data to reconstruct the scene precisely, which aligns with the requirement for more samples in SAR and other active sensing ASP problems.

In all three simple scenes, using a tenth of the viewpoints, the RIFT-100 instance of our RIFT work flow scored up to 247.80% higher in m-SSIM score, up to 15.05% higher in m-COS, up to 68.44% higher in t-IoU, and up to 4.75% lower in p-RMSE across the three simple scene as compared to LS-1000. The detailed data for sphere and pyramid data and figures are available in Appendix A.

### 4.1.2 COMPLICATED SCENE RECONSTRUCTION

For more complex scenes, we constructed two scenarios: "mini parking lot" and "mini highway." In the "mini parking lot" scene, we placed ten "street lights," each 2.2m high, distributed evenly along a line 1.8 m from the y-axis of the scene. There are also two "cars" of different sizes positioned on opposite sides of the road, with dimensions of 0.8m × 0.6m × 0.6m and 1.0m × 0.4m × 0.4m, respectively. We defer the figure and details of the "mini highway" scene to the additional results in Appendix A.

For complex scenes, we used 1,000 viewpoints for the RIFT workflow instances—denoted as RIFT(N)-1000 and RIFT(S)-1000 for the two configurations, respectively—and 2,500 viewpoints for the least squares baseline model, denoted as LS-2500. Using only 40% of the training data, our RIFT-1000 models achieved up to a threefold improvement in m-SSIM score, a 53.79% higher m-COS, and a 567.20% higher t-IoU, although the lead in p-RMSE is smaller. The instances in

Sections 4.1.1 and 4.1.2 where the RIFT model does not perform as well in unseen viewpoint interpolation are likely due to the number of viewpoint samples used.

Table 2: Complicated Reconstruction Result for "Mini Parking Lot" Scene Corresponding to Figure 4

| Model | m-SSIM | m-COS | t-IoU | p-RMSE |
|---|---|---|---|---|
| LS-2500 | 0.1662 | 0.5931 | **0.0356** | 0.0153 |
| RIFT(N)-1000 | 0.5705 | 0.7833 | 0.0244 | 0.0152 |
| RIFT(S)-1000 | **0.6639** | **0.9120** | 0.0345 | **0.0149** |

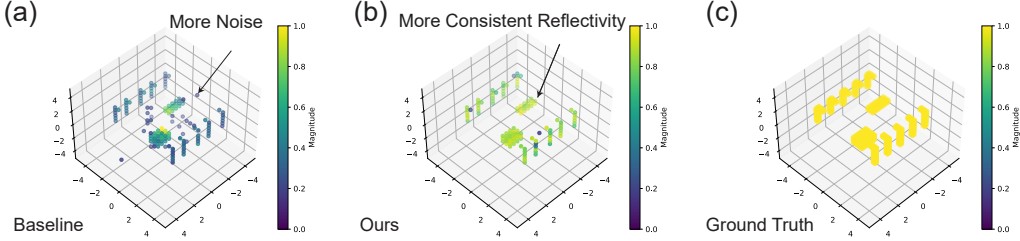

Figure 4: *Visualizations of the "mini parking lot" scene from Section 4: (a) Scene reconstructed by the baseline model. (b) Scene reconstructed by RIFT. (c) Ground truth scene visualized with the same granularity (defined in Section 2.2) as scene reconstruction. Under the same number of input, scene reconstruction by our RIFT model achieved upto 300% higher score in scene reconstruction than baseline by only using 40% of the data samples. The detailed data is in Table 2.*

## 4.2 CASE STUDY: WEAK TARGET DETECTION IN FAR-FIELD

In this section, we investigate a real-world problem in radar signal processing to demonstrate the capabilities of the RIFT. Weak Target Detection (WTD) (Li et al., 2024; Bai et al., 2020) refers to scenarios where multiple objects in a scene have different reflectivities and are positioned close to each other, making it challenging for radar systems to distinguish them.

We use a far-field setting with a smaller scene extent ranging from −3m to 3m and the radar at a distance of 50 m from the scene. The spatial granularity is set to 0.12 m. In the scene, two rectangular reflectors are placed on opposite sides of the y-axis, separated by 1.2m. The dimensions of each bar are 2.4m in length, 0.72m in width, and 0.48 m in height. In order to mimic the practical scenarios, instead of generating signal from a hypothetical sphere surrounding the scene, we limit the azimuth and elevation angle samples to 41 and 21 samples on $[0.1\pi, 0.3\pi]$, respectively.

The bar on the negative x-side is assigned a reflectivity of 1.0, while the bar on the positive x-side is assigned reflectivity of 1.0, 0.5, 0.333, and 0.25 in four separate experiments. Both the RIFT model and the least squares model use 500 input data points. All other training setups are identical to those in the experiments described in Section 4.1. Figure 5 presents the resulting reconstructions of the scene.

From Figure 5(a)–(d), we observe that in far-field simulations, the least squares baseline models are unable to resolve the two reflectors, regardless of differences in their reflectivities. This outcome corresponds to the Weak Target Detection (WTD) problem, where radar systems encountering such scenarios can only identify a general area of reflectivity or may even ignore the weaker object entirely.

In contrast, the RIFT model provides sufficient expressiveness to resolve the two reflectors, even when there is a fourfold difference in reflectivity between them. Although the reconstructed reflectivity of the weaker object is diminished, its accurate localization demonstrates the value of integrating the physical process into the Implicit Neural Representation (INR) in different modalities.

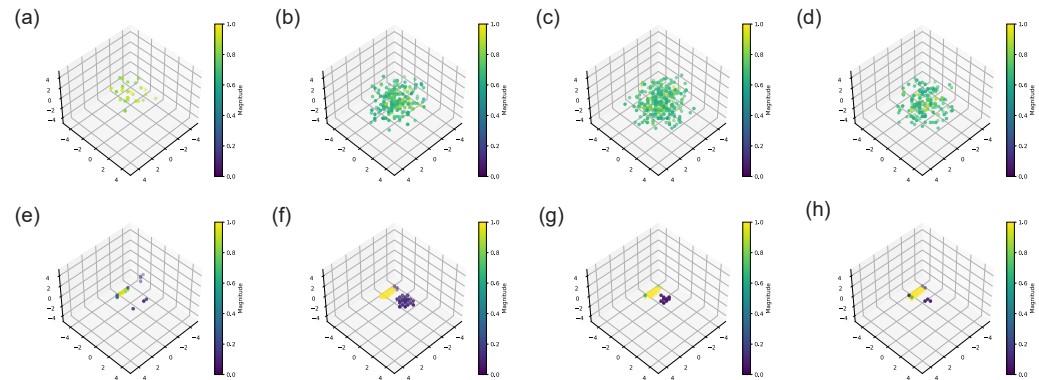

Figure 5: *Visualizations for Weak Target Detection: (a)-(d) Scene reconstruction by the baseline with no difference in reflectivity, 2×, 3× and 4× difference in reflectivity. (a)-(d) Scene reconstruction by the RIFT with no difference in reflectivity, 2×, 3× and 4× difference in reflectivity.*

## 4.3 CASE STUDY: RECONSTRUCTING FROM COMMERCIAL SIMULATOR

In this section, we investigate the difference between the synthetic data and the real world radar signal. We synthesize the radar signal with Ansys Electronics Desktop 2023 R1, referred to as "AEDT",Ans (2024), a ray-tracing based simulation software widely used in different researches as benchmarks including: antenna simulation (Noghanian, 2022), power electronics (Neumaier et al., 2023), magnetics (Bijak et al., 2023), heat management of hardware (Velu et al., 2024).

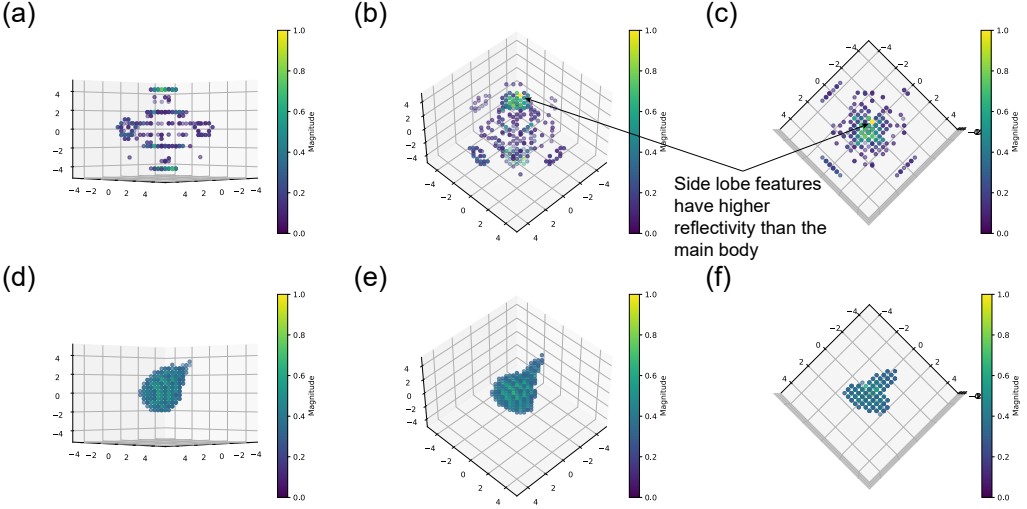

Figure 6: *Visualizations for Reconstruction from Commercial Simulator: (a)-(c) Scene reconstruction by the LS with 1000 data points. (d)-(f) Scene reconstruction by the RIFT 100 data points.*

We reproduced the cube with an edge length of 2m in AEDT. The detailed design parameter for the AEDT is provided in Appendix C. The design is similar to the one is 4.1.1. RIFT uses 100 data points to learn the scene and 20 data point to reconstruct the scene. The LS based reconstruction uses 1000 data points. We can see from the Figure 6, RIFT learns the body of the cubic reflector while the LS picks up more corner and edge features as we expect from normal radar imaging techniques. Additionally, we can see that RIFT does not learn side lobe feature and other radar signal artifacts as LS reconstruction does.

To conclude, we demonstrate the capability of RIFT under real-world data setting considering radar artifacts like side lobes (Yang et al., 2022) and multipath effects(Hao et al., 2022).

## 5 CONCLUSIONS AND FUTURE WORKS

In this paper, we introduced the Radon Implicit Field Transform (RIFT) workflow, which integrates an INR with a traditional forward model for radar signals to reconstruct scenes only from radar data with no exposure to the real scene structure (as compared to (Borts et al., 2024)). Compared to traditional inverse models, RIFT achieves superior scene reconstruction across all experiments and enhances interpolation of unseen viewpoints in certain cases, all while utilizing significantly less data. These results indicate that RIFT effectively addresses the high cost of data acquisition in SAR problems by reconstructing scenes with reduced data requirements.

To assess the performance of RIFT-type models, we introduced customized error metrics for reconstruction and unseen viewpoint interpolation. The m-SSIM empirically aligns with our visual evaluations. However, since RIFT employs a neural network to model scene properties—in contrast to the Kaczmarz-based least square inversion of the forward radar model with well-established convergence properties—it may experience numerical stability issues. Consequently, there is one instance where the RIFT model underperforms in unseen viewpoint interpolation. As illustrated in Figure 7(g), the RIFT model occasionally fails due to convergence to local minima during optimization or vanishing gradients, highlighting the need for further investigation and customized optimization methods. Additionally, data preprocessing techniques like normalization and standardization are not adequate for radar signals because the magnitude after normalization of the 2m cube can differ by orders of magnitude thus undermines the numerical stability.

To fully realize the potential of RIFT, we require datasets of real-world scenes and corresponding radar signals. We reproduce occupancy of simple objects from (Ans, 2024), but further investigation of reconstruction from complicated scene is necessary. We acknowledge the necessity of continued research into RIFT models to bridge them with real-world radar sensing applications, such as compact high-resolution mapping for autonomous vehicles or robotic navigation.

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

## A ADDITIONAL RESULTS

### A.1 ADDITIONAL SCENE RECONSTRUCTION RESULTS AND DETAILS

In this section, we provide the detail of the data of model performances summarized in Section 4. The Table 3 and 4 are for the sphere and pyramid scene discussed in Section 4.1.1.

Due to the nature of radar signal, there are cases when the signal viewing the same scene from a different angle can be apart by orders of magnitude. Hence, there are cases when RIFT training go into local minima or the gradient vanishes, for example, Figure 7 (g). The engineering detail for overcoming the numerical issues are in Appendix B.4 and B.5, but for the sake of completeness, we include the failed experiments in the tables[5].

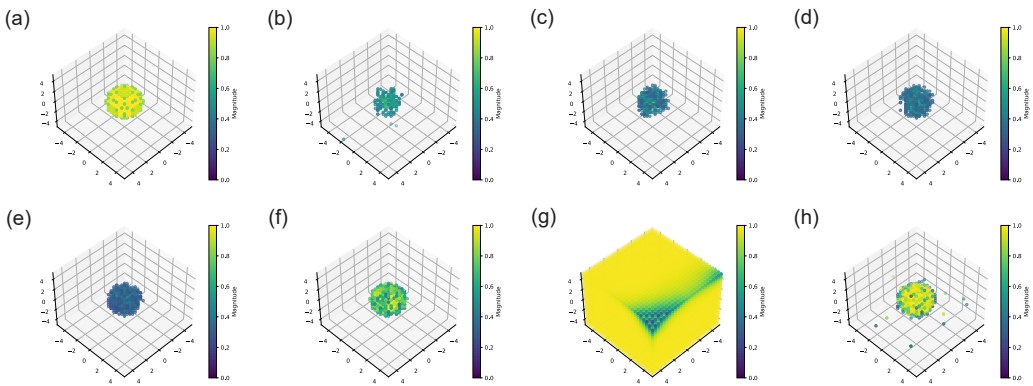

Figure 7: *Visualizations of the "sphere" scene (a): Ground truth of a sphere of radius 2m represented with a granularity of 0.2m. (b)-(e): Scene reconstruction by the baseline with 100, 200, 500, and 1000 viewpoints, respectively. (f): Scene reconstruction by RIFT(N) with 100 viewpoints, respectively. The RIFT-100 scores 188.5% higher in m-SSIM, 11.60% higher in m-COS, and 29.38% higher in t-IoU. However, the p-RMSE lags behind that of LS-1000 by 64.56%. (g),(h): Scene reconstructions by RIFT(N) and RIFT(S) with 1000 viewpoints. The detailed results are presented in Table 3.*

Table 3: Simple Scene Reconstruction Result for Sphere Data in Section 4.1

| Model | m-SSIM | m-COS | t-IoU | p-RMSE |
|---|---|---|---|---|
| LS-100 | 0.0762 | 0.5986 | 0.0897 | 0.0151 |
| LS-200 | 0.1007 | 0.7090 | 0.1741 | 0.0150 |
| LS-500 | 0.1813 | 0.8412 | 0.2130 | 0.0148 |
| LS-1000 | 0.2890 | 0.8886 | 0.2736 | 0.0146 |
| RIFT-100(N) | **0.8343** | **0.9917** | 0.3540 | 0.0187 |
| RIFT-100(S) | 0.2858 | 0.9473 | 0.3412 | 0.0145 |
| RIFT-1000(N)† | 0.0018 | 0.1893 | 0.0893 | 0.0214 |
| RIFT-1000(S) | 0.6002 | 0.9744 | **0.3726** | **0.0145** |

The "mini highway" scene is also in the comprises of a series of "streetlights" positioned 4.4 m from the y-axis of the scene and uniformly spaced by 3.2 m from one another. All "streetlights" are 1.8 m tall. There are "fences" placed 4m from the y-axis and right on the y-axis. The height is 2.0m. There is a 2.4m by 0.8m by 1.6m "car" placed about 3.8m away from the origin of the scene. The Table 5 presents the comparison between the RIFT model and the baseline. This experiment is the only case we noticed a conspicuous disadvantage of the viewpoint interpolation by the RIFT model.

From the results above, we confirm that in all cases we presented, the RIFT models reconstructs the scene better with significantly less data. In most cases, the RIFT models interpolates the unseen viewpoints better with less data. Overall, we demonstrate the potential of the neural representations,

---

[5]In this work, we use †in the tables to denote the failed case we present.

Table 4: Simple Scene Reconstruction Result for Pyramid Data in Section 4.1

| Model | m-SSIM | m-COS | t-IoU | p-RMSE |
|---|---|---|---|---|
| LS-100 | 0.1163 | 0.5584 | 0.0576 | 0.0151 |
| LS-200 | 0.1444 | 0.7012 | 0.1740 | 0.0149 |
| LS-500 | 0.3065 | 0.8158 | 0.1257 | 0.0147 |
| LS-1000 | 0.4258 | 0.8689 | 0.1822 | **0.0146** |
| RIFT(N)-100 | 0.8926 | 0.9840 | **0.2549** | 0.0185 |
| RIFT(S)-100† | 0.0266 | 0.1774 | 0.0581 | 0.0159 |
| RIFT(N)-1000 | **0.9513** | **0.9845** | 0.2537 | 0.0186 |
| RIFT(S)-1000 | 0.6385 | 0.9271 | 0.2730 | 0.0147 |

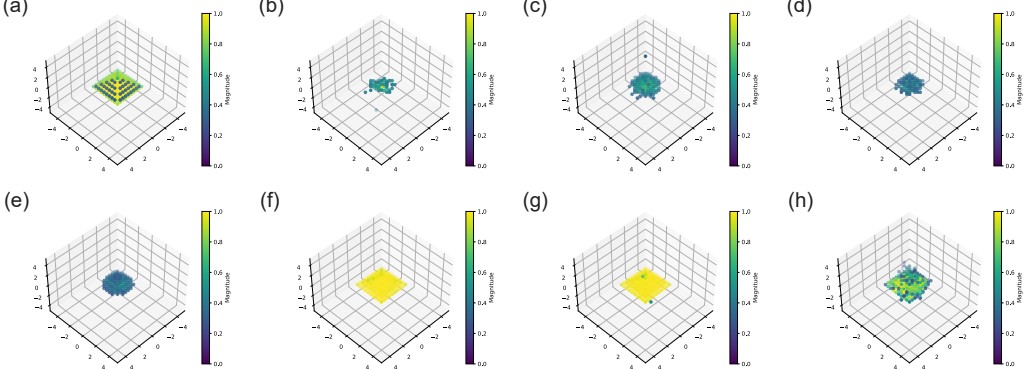

Figure 8: *Visualizations for the "Pyramid" scene(a): Ground truth of a pyramid of cubical base of 2m long and 2m tall represented with a granularity of 0.2m. (b)-(e): Scene reconstruction by the baseline with 100, 200, 500, and 1000 viewpoints, respectively. (f): Scene reconstruction by RIFT(N) with 100 viewpoints, respectively. The RIFT-100 scores 109.60% higher in m-SSIM, 13.25% higher in m-COS, and 39.41% higher in t-IoU. However, the p-RMSE lags behind that of LS-1000 by 59.77%. (g),(h): Scene reconstructions by RIFT(N) and RIFT(S) with 1000 viewpoints. The detailed results are presented in Table 4.*

with a relatively simple model configuration, in the under-researched field. Due to its distinctive nature, radar signal processing may need further investigation on improvement of optimization techniques to prevent the instability (like the one shown in Figure 7 (g)) caused by its wide distribution in magnitude.

## B TECHNICAL DETAILS

In this section, we specify the engineering details in the experiments including details in radar signal processing that are pertinent to this work, the structure of the RIFT models, and the optimization details.

### B.1 RADAR SETUP

In general, we model the radar system as consisting of two components: transmitters (TX) and receivers (RX). We denote the number of TX and RX antennas as $|TX|$ and $|RX|$, respectively. In this study, we set $|TX| = |RX| = 16$. The radar operates over a band of angular frequencies $\omega$, uniformly sampled from $[\omega_{\text{Lo}}, \omega_{\text{Hi}}]$. Here, $\omega$ follows the standard definition in radar signal processing, where $\omega = 2\pi f$ and $f$ is the corresponding frequency. In our synthetic radar simulations, we use 100 frequencies uniformly sampled from the range $[95, 105]$ GHz.

In our problem setup, we define a main three-dimensional coordinate system with its origin at the geometric center of the scene. The radar trajectories for the transmitters and receivers, denoted as $\gamma(s_{TX})$ and $\gamma(s_{RX})$, are determined by an auxiliary trajectory $\gamma_{\text{radar}}$ that lies at a fixed distance

Table 5: Complicated Reconstruction Result for "Mini Highway" Scene Corresponding to Figure 9

| Model | m-SSIM | m-COS | t-IoU | p-RMSE |
|---|---|---|---|---|
| LS-2500 | 0.1914 | 0.5725 | 0.0128 | **0.0154** |
| RIFT(N)-1000 | **0.7354** | 0.8778 | 0.0854 | 0.0160 |
| RIFT(S)-1000 | 0.6897 | **0.9116** | **0.0913** | 0.0160 |

(a)  (b)  (c)

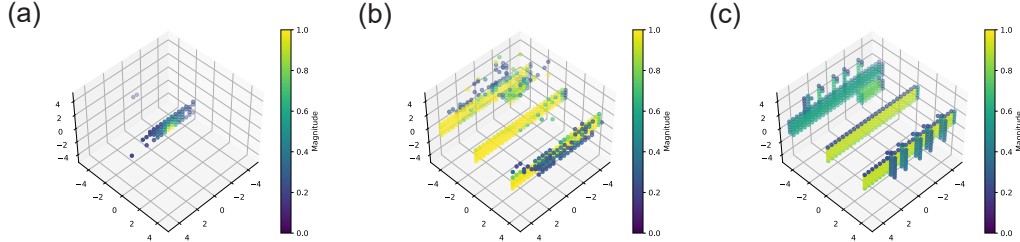

Figure 9: *Visualizations of the "mini highway" scene from Section 4: (a) Scene reconstructed by LS-2500 baseline model instance. (b) Scene reconstructed by RIFT-1000(N) instance. (c) Ground truth scene visualized with same granularity (defined in Section 2.2) as scene reconstruction. The scene reconstruction by RIFT model achieved 243.30% higher score in scene reconstruction and 1.50% better unseen viewpoint interpolation than baseline by only using 40% input data.*

$r_{\text{radar}} = 10\text{m}$ from the origin. To ensure that the radar fully captures the synthetic scene, we assume that the scene's extent, $r_{\text{scene}} = 5\text{m}$, is smaller than $r_{\text{radar}}$. The TX and RX antennas are arranged such that the normal vector of the plane formed by the antennas always points toward the center of the main coordinate system.

The trajectory of $\gamma_{\text{radar}}$ is defined as:

$$\gamma_{\text{radar}} := r_{\text{radar}} \left[\sin\theta\cos\phi, \sin\theta\sin\phi, \cos\theta\right]^T \tag{4}$$

Here, $\theta$ and $\phi$ denote the azimuth and elevation angles relative to the scene center.

For the sake of resolution, we denote the speed of light as $c_0 = 299792458\text{m/s}$ and define the spacing between each individual antenna of the same kind $s = \frac{\lambda_{Max}}{2}$, where $\lambda_{max} = \frac{2\pi c_0}{\omega_{Lo}}$ is the longest wavelength which the radar system uses. In particular, the spacing we use for this work is 1.4276mm. Then for the $m^{th}$ TX and the $n^{th}$ RX, where $m \in [1, |Tx|]$ and $n \in [1, |Rx|]$, the trajectory $\gamma(s_{\text{TX}})$ and $\gamma(s_{\text{RX}})$ are:

$$\gamma(s_{\text{TX}}) = \gamma_{\text{radar}} s(m + \frac{1}{2}) \left[-\cos\theta\cos\phi, -\cos\theta\sin\phi, \sin\theta\right]^T \tag{5}$$

, and

$$\gamma(s_{\text{RX}}) = \gamma_{\text{radar}} s(n + \frac{1}{2}) \left[-\sin\phi, \cos\phi, 0\right]^T \tag{6}$$

respectively.

## B.2 Radar Signal Formulation

In this work, the radar signal $\mathbf{S}$[6] is represented as a three-dimensional complex tensor $\mathbf{S} \in \mathbb{C}^{n_f \times |TX| \times |RX|}$. For each individual experiment, the dimensions of $\mathbf{S}$ are determined by the number of frequencies $n_f$, the number of transmitters $|TX|$, and the number of receivers $|RX|$. Based on the setup described in Appendix B.1, in this work we have $n_f = 100$, $|TX| = 16$, and $|RX| = 16$, so $\mathbf{S} \in \mathbb{C}^{100 \times 16 \times 16}$.

---

[6]In digital signal processing (DSP), $\mathbf{S}$ is often denoted as the S-parameter because it characterizes the scattering properties of a scene.

### B.3 SCATTERING AND ATTENUATION FACTOR

The definition of $\boldsymbol{A}$ comes from the work by Nolan & Cheney (2002):

$$\boldsymbol{A}(\omega, s, \boldsymbol{x}) = \frac{\omega^2 p(\omega) j_s(\omega(\widehat{x - \Gamma(s)}), \Gamma(s)) j_r(\omega(\widehat{x - \Gamma(s)}), \Gamma(s)) m(s)}{4\pi^2 |x - \Gamma(s)|^2} \tag{7}$$

The $\Gamma(s)$ term is the surface which the traces of radar antenna form. The $j_s$ and $j_r$ are Fourier transforms of current density of the radar, which is constant when we fix a radar pattern. The waveform $p(\omega)$ is waveform which we assume to be constant in Appendix B.1. The $m(s)$ term is a taper function which is also constant when we fix the radar. All terms are unity up to a normalization with no loss of information except $|x - \Gamma(s)|^2$.

There are two cases to discuss here: the first being the radar is far enough from the scene, which is often denoted as *far-field* in ASP, the the second being the radar is close to the scene. The difference between the two is that in the far-field case, the difference between the $R_b$ of different combination of TX and RX pair is not significant as compared to the distance which the wave travels. The converse holds true for the *near-field* case.

Consequently, in order to not lose the information from $R_b$, in the case of near-field, the calculation $\frac{1}{R_b^2(s)}$ must be executed before normalization. For far-field, since the $\boldsymbol{x}$ dependence are all constant, the term $\boldsymbol{A}$ is absorbed by the normalization.

That is to say, the GRT of near-field case (which we denote as $\mathcal{F}_{NF}$ below) and far-field case (which we denote as $\mathcal{F}_{FF}$ below) are different where:

$$\mathcal{F}_{NF}[\rho] \approx \int_{\mathbb{R}^3} \frac{1}{R_b^2(s)} e^{j(k(\omega)R_b(\boldsymbol{x}))} \rho(\boldsymbol{x}) e^{\Phi(\rho(\boldsymbol{x}))} d\boldsymbol{x} \tag{8}$$

$$\mathcal{F}_{FF}[\rho] \approx \int_{\mathbb{R}^3} e^{j(k(\omega)R_b(\boldsymbol{x}))} \rho(\boldsymbol{x}) e^{\Phi(\rho(\boldsymbol{x}))} d\boldsymbol{x} \tag{9}$$

Note that in $\mathcal{F}_{FF}$, all $R_b^2 \approx ||\gamma_{radar}||$ for different combinations of TX/RX pairs, and the term is absorbed by normalization. From the radar setup in Appendix B.1, we use $\mathcal{F}$ as a shorthand $\mathcal{F}_{NF}$ for this study unless noted otherwise.

### B.4 MODEL STRUCTURE

The two configurations of the RIFT models share the same structure shown in Figure 10. Both models have 3 dimensional input of the radar array center position $\gamma_{\text{radar}}$. The output of the models are 2 dimensional, which are real and imaginary part of learned scene reflectivity $\hat{\rho}(\boldsymbol{x})$. The hidden size of all hidden layers are 64.

The primary difference between the RIFT(N) and RIFT(S) models arises from the definitions of their units and nonlinearities. In the RIFT(N) models, each unit consists of a Linear layer, followed by a nonlinearity $\sigma$, and then a LayerNorm layer. The nonlinearities $\sigma$ and $\tau$ are the LeakyReLU function (Maas et al., 2013) and the hyperbolic tangent function, respectively. In contrast, the RIFT(S) models have a simpler structure: each unit is a single Linear layer, and all nonlinearities $\sigma$ and $\tau$ are sine functions.

As previously mentioned, the optimization process faces challenges due to the dynamic range of radar signals. Among all the different multilayer perceptron (MLP) structures we experimented with, the INR architectures used in RIFT(N) and RIFT(S) models were empirically found to perform the best. These structures optimize effectively despite frequent vanishing gradients and convergence to local minima.

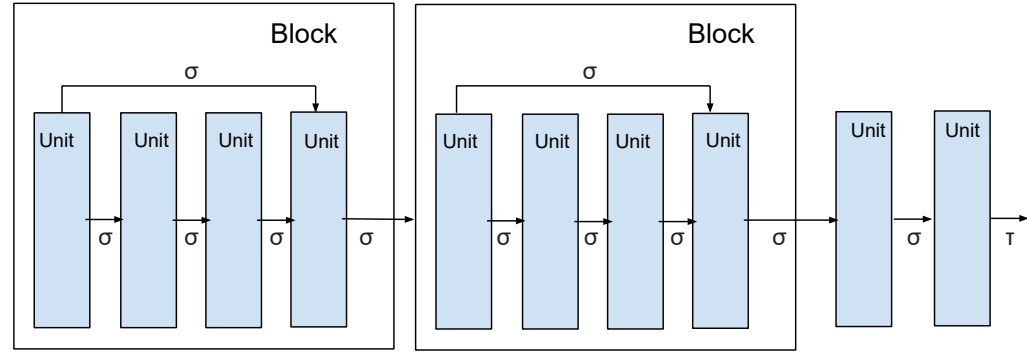

Figure 10: *The common architecture of two different configurations of RIFT models.*

### B.5 OPTIMIZATION DETAILS

In practice, we observe that the optimization problem Equation 3:

$$\arg\min_{\Theta} ||\mathbf{S} - \mathcal{F}[\hat{\rho}_\theta(\boldsymbol{x})]||_2$$

often faces challenges due to vanishing gradients. We speculate that since PyTorch's autodifferentiation Paszke et al. (2019) implements numerical calculations of differences in function values, the high dynamic range of radar signals can lead to computations involving very small numbers. Additionally, these small numbers are scattered across all frequencies and combinations of TX and RX, which further ill-conditions the loss landscape. To avoid multiplication of small numbers that could compromise numerical stability during training, we instead minimize the $L_1$ norm:

$$\arg\min_{\Theta} ||\mathbf{S} - \mathcal{F}[\hat{\rho}_\theta(\boldsymbol{x})]||_1 \tag{10}$$

Our results support this approach.

For all experiments presented in this paper, we used the AdamW optimizer (Loshchilov & Hutter, 2018) with an initial learning rate of $10^{-2}$ and a weight decay rate of $10^{-2}$. Optimization was set to cease after 500 epochs, with learning rate annealing (You et al., 2023) by halving every 100 epochs. Throughout this work, we handle the magnitude and phase of radar signals separately. We take the results with the lowest loss during the 500 epochs as the final result. During training, we process the real and imaginary parts of the loss function separately and assign them different weights because the magnitude of the radar signal spans the radar's dynamic range, whereas the phase is confined to $[0, 2\pi]$. In all experiments presented, the weight assigned to the phase term in the loss function is typically several thousand times greater than that of the magnitude term.

For the baseline models, all least square solution are solved with 100 iterations of block-Kaczmarz (Kaczmarz, 1993) algorithm.

### B.6 FURTHER DISCUSSION ON METRICS

The mathematical intuition behind separating SSIM for magnitude and phase stems from the formulation of the INR (see Section 3.2) and forward radar signal synthesis. In radar signal synthesis, we assign a reflectivity function $\rho(\mathbf{x})$ to the scene, as discussed in Section 2.2. For simplicity, when the scene is not occupied by a particular object, the reflectivity is set to zero. The subset $\mathbf{x}_0$, where $\rho(\mathbf{x}_0) = 0$, corresponds to regions of the scene without objects, while the subset $\mathbf{x} \setminus \mathbf{x}_0$ represents the parts of the scene occupied by objects.

By comparing the geometry of the occupied regions $\mathbf{x} \setminus \mathbf{x}_0$ with the ground truth geometry, we can assess how well the INR learns the scene, even at angles where reflections from the scene are weak. In all the experiments presented in this work, we select 100 unseen viewpoints and calculate the p-RMSE of all radar signals across these viewpoints.

The threshold Intersection-over-Union (tIoU) metric is inspired by the constant false alarm rate (CFAR) used in radar signal processing. In radar imaging, persistent background noise is present in radar signals, and various methods have been developed to reduce the influence of this constant false alarm (Schou et al., 2003; Hou et al., 2015). In this work, to address CFAR, we introduce a threshold to the magnitude of the learned reflectivity $\hat{\rho}(\mathbf{x})$ by assuming an apparent SNR, and discard all values below this threshold. We then calculate the Intersection-over-Union (IoU) to assess how much of the scene has been learned, especially in the reconstruction of a single object.

Due to the unavoidable noise level, tIoU values are often low because the presumed SNR cutoff must be held constant when comparing reconstructions from different methods, and it cannot be optimal for all methods simultaneously. However, tIoU can be considered a robustness measure for the model, since under the same presumed threshold, a higher tIoU value indicates better alignment between the ground truth scene and the reconstructed scene. The less noise present in the model's inference, the more of the inference surpasses the presumed SNR threshold, resulting in a higher tIoU.

For the calculation of tIoU and figure generation, we apply the thresholding of the presumed SNR at a fixed value for a fixed number of viewpoints. The thresholding only affects the results of tIoU and visualization. The m-SSIM calculations are conducted by slicing the 3D scene on a fixed 2D plane, and the result for a scene is the average m-SSIM of all slices.

For the unseen viewpoint interpolation, we measure the Mean Squared Error (MSE) of the phase of the radar signal, which we denote as p-RMSE. We discard the magnitude in the MSE calculation due to the nature of radar signals. Given a scene, reflections from certain viewpoints can differ by orders of magnitude if the reflectivity $\rho(\mathbf{x})$ is anisotropic. To test the generalizability of the model, we aim to reduce the impact of the signal magnitude, focusing instead on the phase information, which is necessary for the correct coherent addition of radar signals in SAR imaging.

### B.7 FURTHER DISCUSSION ON HYPERPARAMETER TUNING

As introduced in Section 3.3, in the training process, the error of the radar signal generated by GRT is divided to two parts: the magnitude and the phase. Consequently, the loss function in the actual optimization mentioned in B.5 can be written as:

$$\alpha_1 ||\,||\mathbf{S}|| - ||\mathcal{F}[\hat{\rho}_\theta(\boldsymbol{x})]||\,||_1 + \alpha_2 ||\widehat{\mathbf{S}} - \widehat{\mathcal{F}[\hat{\rho}_\theta(\boldsymbol{x})]}||_1 \tag{11}$$

Here, $\widehat{\cdot}$ refers to the phase angle of a complex signal. $\alpha_1$ and $\alpha_2$ are the weight term for the loss term related to the error in magnitude and magnitude in phase.

Among all parameters used in this work, $\alpha_1$ and $\alpha_2$ is a pair of parameter of the most importance. We know that the coherent addition of the signal is the principle of radar imaging (Woodman, 1997). So the correct addition of the phase information is a condition of correct imaging. Typically, $\alpha_2$ is set to be $1000\times$ of $\alpha_1$. Under different circumstances, the ratio $\frac{\alpha_1}{\alpha_2}$ should be adjusted to find the balance between the parameters so that RIFT can learn the representation of the scene.

## C DESIGN PARAMETER USED IN SECTION 4.3

In Section 4.3, we used Ansys Electronics Desktop 2023 R1 (referred to as "AEDT") as a simulation tool to produce radar signals. We use the "shooting and bouncing rays" (SBR) mode of the software. In the SBR simulation, we attempt to reproduce the exact experiment with our data synthesis3.1. However, the bandwidth is automatically calculated. We can only provide the setup of the SBR simulation is in the following Table 6 and 7:

The velocity of the radar does not matter since the scene is static. However, for the sake of the completeness of the setup, we set the velocity resolution to be 0.4m/s and the min-max velocities to

Table 6: AEDT SBR Simulation Type Setup

| Setup Type | Waveform Type | Channel Configuration |
|---|---|---|
| Range Doppler | Chirp-Sequence FMCW | I+Q Channels |

Table 7: AEDT SBR Range-Doppler Configuration

| Center Frequency | A/D Sampling Rate | Range Resolution | Range Period |
|---|---|---|---|
| 100 GHz | 10 MHz | 0.015m | 1.5m |

be $\pm 2$m/s. The resulting waveform parameters calculated by AEDT are:

$$\text{Radar Bandwidth: } 9993.082\text{MHz}$$
$$\text{Chirp Duration: } 10\mu s$$
$$\text{Frequency Ramp Rate: } 999.308\text{MHz}/\mu s$$
$$\text{Number of A/D Samples per Chirp: } 100$$
$$\text{Coherent Processing Interval Duration: } 3.747\text{ms}$$
$$\text{Coherent Processing Interval Number of Chirps: } 10$$
$$\text{Pulse Repetition Frequency: } 2.669\text{KHz}$$
$$\text{Chirp Duty Cycle } 2.668\%$$

Notice that the calculated radar bandwidth is not exactly the same as the setup in B.1. Consequently, we are unable to directly calculate the difference between our synthesized data and the simulated data from AEDT.

For this experiment, the radar is located at 10m distance from the center of the scene. The radar setup is identical to the one in Appendix B.1 except that the sampled viewpoints are generated from 41 uniformly sampled azimuth and elevation angles from $[0, 2\pi]$ and $[0, \pi]$, for a total of 1681 data points. We reduce the number of generated data only because of computational resources and this reduction does not change the experiments.

