# OpenReview forum: "Radon Implicit Field Transform (RIFT): Learning Scenes from Radar Signals"
_ICLR.cc/2025/Conference — Submitted to ICLR 2025_

### Official Review · Reviewer_zohP · 2024-10-28

**Soundness:** 3
**Presentation:** 3
**Contribution:** 4
**Rating:** 3
**Confidence:** 4

**Summary:**

This paper presents RIFT (Radon Implicit Field Transform), a novel method for learning scene representations directly from radar signals using implicit neural representations (INRs). The approach combines classical radar forward modeling (Generalized Radon Transform, GRT) with modern neural network techniques to address the data acquisition challenge in array signal processing (ASP), particularly in Synthetic Aperture Radar (SAR) applications.

The main contributions include:

1. A framework that integrates INR with GRT to learn scene representations directly from radar signals, enabling scene reconstruction with reduced data requirements

3. Introduction of new evaluation metrics specifically designed for radar signal processing:
   - phase-Root Mean Square Error (p-RMSE) for radar signal interpolation
   - magnitude-Structural Similarity Index Measure (m-SSIM) for scene reconstruction

The authors demonstrate their method's effectiveness through extensive experiments on synthetic data, showing improvements in both scene reconstruction quality and viewpoint interpolation capability compared to traditional methods. They also present a case study on weak target detection in far-field scenarios, demonstrating the method's potential practical applications.

**Strengths:**

Good Originality:

- The paper presents a novel application of implicit neural representations to radar signal processing, particularly in SAR imaging
- While similar to computer vision approaches, its adaptation to radar domain represents meaningful innovation

Good Quality:

- The technical development is thorough and well-grounded in both radar signal processing and deep learning principles
- The paper introduces well-designed evaluation metrics specifically for radar signals:
  * phase-Root Mean Square Error (p-RMSE) for signal interpolation
  * magnitude-Structural Similarity Index Measure (m-SSIM) for scene reconstruction

**Weaknesses:**

# Major Weaknesses:

1. **Lack of Real-World Data Validation**: **The most critical concern**
- The paper relies entirely on synthetic data for evaluation
- Many public mmWave and SAR datasets are available but not utilized
- The gap between synthetic and real data is not addressed:
  * Real scenarios involve complex multipath effects
  * The accuracy of GRT in real-world conditions is questionable
  * Environmental factors affecting radar signals are not considered
- The effectiveness on synthetic data is expected due to neural rendering, but this doesn't validate real-world applicability
- Therefore, this significant omission raises serious doubts about the validation and practical value of the proposed method

> I would significantly improve my assessment if the authors could demonstrate their method's effectiveness on real-world SAR or mmWave radar datasets.

2. **Accuracy of GRT Forward Model**
- The paper doesn't address GRT's limitations in complex real-world scenarios:
  * Multipath effects
  * Material properties
  * Environmental interference
- These limitations could severely impact the method's practical applicability
- The validation of GRT accuracy is crucial but missing from the current work

3. **Related Work Citiation and Discussion**
- RadarFields (published May 2024) has already explored similar ideas
- While concurrent development is understandable, proper citation and discussion are needed

# Minor Weaknesses:

- Inconsistent mathematical notation in Section 3.1:
  * Variable x is used inconsistently
  * Matrix A notation varies throughout the paper

- Typos

**Questions:**

Please see weaknesses

---

> ### Author Response · Authors · 2024-11-23
> **Reply to Reviewer zohP**
>
> Dear Reviewer zohP,
>
> We sincerely appreciate your review and valuable feedback.
>
> Based on the weaknesses you pointed out, we made changes to the manuscript to improve its overall quality. Specifically, to address the #1 major weakness you mentioned, we included a new Section 4.3 in the updated manuscript.
>
> Here is a point-by-point response to the weaknesses you mentioned and the efforts we made to address them:
>
> 1. Dataset Applicability:
> The mmWave and SAR datasets we found are not applicable to this problem. For example, the commonly used mSTAR dataset (https://www.kaggle.com/datasets/atreyamajumdar/mstar-dataset-8-classes) contains SAR images but not the radar signals used to generate the images. In the meantime, we used commercial simulation software based on ray tracing, Ansys Electronics Desktop (referred to as “AEDT”), which has been used in many research studies as a benchmark (e.g., Phan et al., 2023; Ren et al., 2017). This is as close to real-world data as we can get with simulations.
>
> We reconstructed a cube (provided the resolution is not high) using AEDT-generated data and provided the details in the newly added Section 4.3.
> We learned the AEDT-generated scene using GRT-based RIFT. Although the scene was not complex, this partially addresses your concerns about the gap between fully synthetic data and reality.
> We are working on reconstructing more complicated scenes and hope to provide an update early next week.
>
> 2. Accuracy of the GRT Model:
>
> The accuracy of the GRT model is partially demonstrated in the case studies. For example, although the “true signal” in Section 4.2 was generated by our forward model, we tested two objects of the same shape with different reflectivities. RIFT was able to resolve the two objects, whereas the traditional method could not.
> The new Section 4.3 can serve, at least partially, as a validation tool for RIFT under real-world conditions.
>
> 3. Discussion of Related Work:
>
> We updated the manuscript to address the lack of discussion you highlighted. We acknowledge that we were not aware of RadarField at the time of composing the first draft of this work.
>
> 4. Typographical and Notation Errors:
>
> In the updated manuscript, we fixed all the typos and inconsistent notations you mentioned.
> We hope that our responses address your concerns and contribute to a potential improvement in your evaluation of our work.
>
> Best regards,
> Authors of Submission 12994
>
> References:
> 1. Tra Nguyen Phan, Bengt Oelmann, and Sebastian Bader. 2023. Towards Automated Design Optimization of Electromagnetic Energy Harvesting Transducers. In Proceedings of the 20th ACM Conference on Embedded Networked Sensor Systems (SenSys '22). Association for Computing Machinery, New York, NY, USA, 871–877. https://doi.org/10.1145/3560905.3568102
>
> 2. Ren, Q., Nagar, J., Kang, L., et al. Efficient Wideband Numerical Simulations for Nanostructures Employing a Drude-Critical Points (DCP) Dispersive Model. Scientific Reports, 7(2126), 2017. https://doi.org/10.1038/s41598-017-02194-1

---

> > ### Comment · Reviewer_zohP · 2024-11-24
> >
> > Thank you for the comprehensive response and the considerable effort in addressing the concerns. The added AEDT simulation is a step forward and shows RIFT's potential in controlled environments.
> >
> > However, I must emphasize that simulation results, including AEDT, cannot substitute for real-world validation. While I understand the challenges, there are several public datasets with raw radar data (such as Coloradar) that would be suitable for validating RIFT. Real-world validation is essential because it would demonstrate RIFT's robustness to practical challenges like environmental noise, multipath effects, and hardware limitations - factors that simulations fundamentally cannot fully replicate.
> >
> > Given the availability of public radar datasets, I believe demonstrating RIFT's effectiveness on actual radar data is crucial for establishing its practical value. Without such validation, it's difficult to assess RIFT's real-world applicability, regardless of how well it performs in simulations.

---

> ### Author Response · Authors · 2024-11-26
> **Reply to Reviewer zohP**
>
> Dear Reviewer zohP:
>
> We appreciate your prompt response. While we agree that simulation is not a substitute for the real world, we believe that it is not fair to fully discount the successful reconstruction of scenes from commercial simulation softwares.
> To support our argument, we want to provide some information regarding the tool we use and the difference between scene reconstruction based on visual and radar signals.
>
> 1. The particular tool we used is widely accepted as an industry standard in antenna design and validation before manufacturing for at least a decade (Sun et al., 2014, Xu et al., 2019, Blair et al., 2019).
>
> 2. We are sorry that we did not directly mention this in the previous response, but we have provided in the general note that the tool we use does address multipath effect with ray tracing (please refer to this article for more information: https://www.ansys.com/blog/using-synthetic-data-for-radar-applications).
>
> 3. The most important noises to radar signals are thermal noise and clutter. Addition of Gaussian noise can correctly mimic behavior of thermal noise in radar (Rouphael, 2014). We will provide the result with thermal noise in the revision. There are other tools from the same company used in modeling for autonomous vehicles that address clutter and they use it with an automobile manufacturer (https://www.ansys.com/it-it/campaigns/bmw).
>
>
> Given the usage of Ansys by both antenna designers and automotive companies, we are confident that it is suitable for this work.
> We would also like to argue in favor of our work in a more tangible way by illustrating the difference between 3D reconstruction with visual data and with radar signals.
>
>
> 1. As it’s known to the ML community, hardware limitation is a concern for all works. What we try to achieve in this work is to emulate a synthetic aperture radar with implicit neural representation. The amount of data needed to achieve synthetic aperture radar imaging is significantly greater than that of optical imaging. To resolve 2 features spaced by 0.01m in a scene of dimension 10m with no aliasing, the number of "colors" or frequency samples required is 1000. The demand for data in radar imaging is significantly higher than that in optical imaging. The non-INR based reconstruction methods used significantly larger network while they are still using point clouds (Sun et al. 2021). We note that given the large amount of training data, our model achieves a large order of compression (>1000x)
>
> 2. The applicability of off-the-shelf dataset: While we appreciate the dataset you mentioned, it is not fully applicable to our problem. For the coloradar (Kramer et al. 2022), in particular, while the moving radar in a scene can be resolved with a change in the reference coordinate, the frame rate is 10Hz and 5Hz. We require both densely sampled viewpoints over a wide aperture, ideally with a stationary scene. While we do believe we could use this dataset eventually, it would require significant data cleaning and calibration which is not suitable for the revision. We hope our argument for the Ansys high-fidelity industry standard simulation will suffice.
>
> Best Regards,
> Authors of Submission 12994
>
>
> References:
> 1. C. Sun, H. Zheng, L. Zhang and Y. Liu, "Analysis and Design of a Novel Coupled Shorting Strip for Compact Patch Antenna With Bandwidth Enhancement," in IEEE Antennas and Wireless Propagation Letters, vol. 13, pp. 1477-1481, 2014, doi: 10.1109/LAWP.2014.2341596.
> 2. X. Xu et al., "Intelligent Design of Reconfigurable Microstrip Antenna Based on Adaptive Immune Annealing Algorithm," in IEEE Transactions on Instrumentation and Measurement, vol. 71, pp. 1-14, 2022, Art no. 8002714, doi: 10.1109/TIM.2022.3162281.
> 3. C. Blair, S. López Ruiz and M. Morales, "5G, A MultiPhysics Simulation Vision From Antenna Element Design to Systems Link Analysis," 2019 International Conference on Electromagnetics in Advanced Applications (ICEAA), Granada, Spain, 2019, pp. 1420-1422, doi: 10.1109/ICEAA.2019.8879074.
> 4. Tony J. Rouphael, Chapter 3 - Noise in Wireless Receiver Systems, Wireless Receiver Architectures and Design, Academic Press, 2014, Pages 105-178, ISBN 9780123786401, https://doi.org/10.1016/B978-0-12-378640-1.00003-2.
> 5. Y. Sun, Z. Huang, H. Zhang, Z. Cao and D. Xu, "3DRIMR: 3D Reconstruction and Imaging via mmWave Radar based on Deep Learning," in 2021 IEEE International Performance, Computing, and Communications Conference (IPCCC), Austin, TX, USA, 2021, pp. 1-8, doi: 10.1109/IPCCC51483.2021.9679394.
> 6. Kramer, Andrew, Kyle Harlow, Christopher Williams, and Christoffer Heckman. “ColoRadar: The direct 3D millimeter wave radar dataset.” The International Journal of Robotics Research 41, no. 4 (2022): 351-360.
>
> ---
> Edit: replaced an invalid link

---

> > ### Comment · Reviewer_zohP · 2024-11-28
> >
> > Thank you for your detailed and thoughtful response. I appreciate the considerable effort you have made to address the concerns, particularly through the addition of AEDT simulations and the comprehensive explanation of its industry-standard capabilities.
> >
> > While I acknowledge that ANSYS is indeed a highly sophisticated simulation tool widely used in industry and research, I maintain my position regarding the fundamental importance of real-world data validation. Here's why:
> >
> > 1. Even in computer vision, where simulation and rendering have reached remarkable levels of sophistication, there remains a notable gap between synthetic and real data. This gap is generally more pronounced in radar applications due to:
> >    - Complex environmental interactions
> >    - Material-dependent reflection characteristics
> >    - Multipath effects
> >    - System-level hardware imperfections
> >    - Environmental noise and interference
> >
> > 2. The radar community has established a strong consensus regarding real-world data validation. In academia, virtually all top-tier publications across ML, wireless sensing, and radar venues consider real data validation an essential requirement. One of the most significant challenges researchers face is successfully bridging the gap between simulation and real-world performance. Similarly in industry, while sophisticated simulation tools like ANSYS serve as invaluable resources for initial algorithm development and verification, the ultimate validation invariably relies on real-world experiments.
> >
> > 3. Therefore, while ANSYS is impressive, they represent idealized approximations of real-world phenomena. The true test of an algorithm's robustness lies in its ability to handle the unpredictable variations and imperfections present in real-world scenarios.
> >
> > In conclusion, I would be inclined to give a much higher evaluation (6+) if RIFT could demonstrate effectiveness with real data (whether through zero-shot, few-shot, or retraining), as I find RIFT's core idea promising. However, without real-world validation, RIFT lacks convincing validation and impact. Therefore, I maintain my original rating.

---

### Official Review · Reviewer_B7gc · 2024-11-01

**Soundness:** 2
**Presentation:** 2
**Contribution:** 3
**Rating:** 5
**Confidence:** 3

**Summary:**

The paper describes an application of implicit neural representation with a radar forward model to recover the 3D complex reflectivity of a scene radar array observations. A bi-static forward model is introduced and used to supervise a SIREN network with recovery results shown on simulated test cases. Two evaluation metrics are designed to quantify success both in reflectivity reconstruction and novel radar view synthesis.

**Strengths:**

The strength of the paper is the introduction of a new forward model to the ICLR community and its integration with neural fields. In general this is an interesting inverse problem which could a) open up new areas for research with the ML community and b) use ideas from the ML community, such as neural fields, for improving reconstructions, uncertainty quantification, run-time and scalability etc.

**Weaknesses:**

Overall I like the idea of the paper, however, I think the execution is quite lacking and it is not quite ready for publication. The experimental design is quite simplistic and I am unable to verify basic things like the validity of the forward model or if any improvements from the inclusion of neural fields would translate to real world scenarios. Below I detail some of the bigger issues and some minor ones:

1. The main issue is that the simulations are way too simplistic. SAR is a well established imaging modality and so I would expect to see real world data reconstructions. In the absence of real data reconstructions I would still expect to see more realistic simulations than the ones shown in the paper. I don’t think we can learn anything from the cube experiments and how they translate to real world scenarios.

2. The experimental setup is not adequately described, so it is hard to assess the methodology. Here is a small subset of questions to guide the authors to a more complete picture:
  2.1. What is the sampling of the view angles? Is it sampled uniformly? Is it sampled according to a realistic path of a SAR instrument?
  2.2. What noise, if any, is added to the data? Is this noise realistic and mimics real SAR instruments?
  2.3. Is there any mismatch between the forward GRT model used to generate the observations and the one used to recover the reflectivities?
  2.4. Is the data/radar assumed to have perfect calibration, is this assumption realistic?
  2.5. Is the GRT model described in Line 146 adequate? Are there any underlying assumptions that would introduce noise (systematic or otherwise) to real radar observations? What is the noise model?
  2.6. How does R_b (Line 149-150) take into account differences in transverse vs along line of sight movement of the SAR?
  2.7. In sec 3.1 it is not clear what is granularity refers to when SIREN is a continuous representation
  2.8. The entire paragraph and discussion given in lines 188-193 is completely unclear. Also matrix A was never defined in the text.
3. I think that there are some over claims in the text that should be addressed.
  3.1 For example lines 216-217 state: “It is crucial to note that during training, we employ a nonstandard approach of accumulating gradients within an individual epoch across different viewpoints.” . This seems to me quite standard in multi-view setting, and I think even the original NeRF and any multi-view algorithm before nerf (e.g. bundle adjustment, multi view stereo, etc) uses rays from multiple views in each gradient calculation. Furthermore the highlighted claim: “This gradient accumulation is specifically designed to mimic the physical motion inherent in synthetic aperture radar systems” is not entirely clear to me, how is that different to any other multi-view setup? Where is the motion taken explicitly into account for the SAR system. It would help the paper if you could explicitly clarify these statements that seem like a key contribution.
  3.2. Line 485: “lays a cornerstone for research into the representation of INRs in less-explored data modalities.” I think this is an over claim. This work certainly is in line with a recent trend but I wouldn’t say it is laying the corner stone. Work in the past couple of years have demonstrated the use of nerf with wild and interesting forward models from biology, astronomy, transient imaging to name a few, see a handful of links below:
            - https://arxiv.org/abs/2307.09555
		- https://arxiv.org/abs/1909.05215
            - https://arxiv.org/abs/2204.03715
           - https://arxiv.org/abs/2405.04662
           - https://arxiv.org/abs/2309.04437
           - https://ojs.aaai.org/index.php/AAAI/article/view/20171


Minor comments:
 - Figure 1 should not appear in the first page where it’s not referenced.
 - Could be useful to give equation numbers for referencing.
 - The use of “ground truth” should be reserved for reflectivities and it is confusing to use it in the context of radar data, even if this “data” was simulated from ground truth reflectivities. (e.g. lines 173, 231 etc).
 - Lines 418-424 describe the setup in a very complex way. I think this could be resolved with a single illustration figure.

**Questions:**

See weaknesses.

---

> ### Author Response · Authors · 2024-11-23
> **Reply to Reviewer B7gc**
>
> Dear Reviewer B7gc,
>
> We sincerely appreciate your review and valuable feedback.
>
> Based on the weaknesses you pointed out, we made changes to the manuscript to improve its overall quality and credibility.
>
> Here are our efforts to address the weaknesses you mentioned:
>
> 1. We used commercial simulation software “Ansys Electronics Desktop 2023 R1” (referred to as “AEDT”), which is based on ray tracing and incorporates most of the effects encountered in real radar imaging tasks. In the updated manuscript, we provided details in the new Section 4.3. While this still does not fully address weakness #1 you pointed out, we are working on reconstructing more complex scenes using data from ray tracing simulations that are as close to real-world data as possible. We hope the reconstruction of scenes using commercial simulation software can enhance the credibility of our work, at least partially bridging the gap between synthetic and real-world data.
>
> 2. In the new manuscript, we further elaborated on the experimental setup in Appendices B.2, B.3, and B.7.
>
> 3. Several details addressing questions in weakness #2 appear in different sections. For example, the sampling of view angles is described in lines 299 to 304 of the previous submission.
>
> 4. There are indeed some questions you mentioned in weakness #2 that we did not address well. Below, we provide point-by-point answers to these questions so you don’t have to search the updated manuscript for the answers.
>
> We also hope the updated manuscript addresses your concerns more comprehensively.
>
> We conducted further investigation into related work and improved the writing. We hope the rewritten sections, which are described in the general response to all reviewers, address this concern.
>
> Point-by-Point Answers to Questions in Weakness #2:
> 1. Sampling of View Angles:
> The sampling of view angles is described in lines 299 to 304 of the previous submission. We acknowledge that these details could have been placed differently. The problem setups corresponding to Sections 4.1, 4.2, and the newly added Section 4.3 are distinct, so they were not included in Section 3. Accordingly, we renamed Section 4 to “Experimental Setups, Results, and Discussions.”
>
> 2. Matrix A:
> We acknowledge that we did not include a link to Appendix B.3 at its first mention. Matrix A is defined in B.3, and the link to B.3 previously appeared at the end of Section 3.1 (lines 192–193).
>
> 3. Noise in Radar Data:
> We did not add any noise to the radar data. The GRT itself is a first-order approximation (under the Born approximation, which simplifies interactions between the scene and the wave to interactions between each voxel and the wave) and inherently contains inaccuracies, such as not accounting for the multipath effect mentioned by reviewer zohP. We updated the manuscript accordingly. However, we acknowledge that the absence of noise could impact your evaluation of our work's robustness. We are running experiments with Gaussian noise and will update the manuscript next week.
>
> 4. Choice of Forward Model:
> We chose the GRT as the forward model of radar, which is not directly related to radar calibration. The ability to normalize the radar signal's amplitude in our problem setting is explained in Appendix B.3.
>
> 5. GRT Model:
> The GRT model operates under the Born approximation, as stated in Section 3. Other approximations, such as the Kirchhoff approximation, also simplify Maxwell’s equations, which are the accepted "adequate" model for electromagnetic waves. The imperfections in reconstructions from AEDT simulations can partially be attributed to these approximations. However, GRT and its variants are widely used in modeling wave propagation, especially in radar signal processing (e.g., Xia et al., 2016; Ding et al., 2019; Zhang and Pi, 2013).
>
> 6. Voxel and Radar Position Tracking:
> The positions of voxels and the radar can be tracked because they share a unified coordinate system.
>
> 7. Granularity:
> Granularity refers to the resolution used to discretize the scene and prepare the data. It works the same way for SIREN and the normalized version of RIFT. We clarified this point in the updated manuscript in the second paragraph of Section 3.1.
>
> 8. Details of Lines 188–193 in the Previous Submission:
> The details corresponding to lines 188–193 in the previous submission are in Appendix B.3. Due to the 10-page limit, we were unable to include this explanation in the body of the paper.
>
> We hope our responses address your concerns and contribute to a potential improvement in your evaluation of our work.
>
> Best regards,
> Authors of Submission 12994
> (Removed the reference for character limit, will provide in the next reply.)

---

> ### Author Response · Authors · 2024-11-23
> **Reference for the previous reply**
>
> References:
> 1. Xia, W., Zhou, Y., Jin, X., & Zhou, J. (2016). A Fast Algorithm of Generalized Radon-Fourier Transform for Weak Maneuvering Target Detection. International Journal of Antennas and Propagation, 2016(1), 4315616. https://doi.org/10.1155/2016/4315616
>
> 2. Z. Ding et al., "A Ship ISAR Imaging Algorithm Based on Generalized Radon-Fourier Transform With Low SNR," IEEE Transactions on Geoscience and Remote Sensing, vol. 57, no. 9, pp. 6385-6396, Sept. 2019, doi: 10.1109/TGRS.2019.2905863.
>
> 3. Biao Zhang and Yiming Pi. (2013). A 3D Imaging Technique for Circular Radar Sensor Networks Based on Radon Transform. International Journal of Sensor Networks, 13(4), 199–207. https://doi.org/10.1504/IJSNET.2013.055582

---

> > ### Comment · Reviewer_B7gc · 2024-11-27
> >
> > I appreciate the revisions and thorough responses by the authors. I have read the responses, the other reviews, and the new material added. I think the new paper is improved by the more realistic simulations that include ray tracing simulations. Nonetheless, while the inclusion of realistic radar signal simulation is an important step in the right direction, I still maintain that the scene itself appears to be very simple, consisting of one single elongated blob. I acknowledge that I have no experience with radar observations, and I am willing to update my rating. That being said, I still remain on the side of reject due to the simlicity of the simulated and reconstructed structure.

---

### Official Review · Reviewer_dWXL · 2024-11-02

**Soundness:** 3
**Presentation:** 3
**Contribution:** 2
**Rating:** 6
**Confidence:** 4

**Summary:**

This paper addresses the challenges of data acquisition in array signal processing (ASP), particularly in Synthetic Aperture Radar (SAR), where high angular and range resolutions require extensive data collection. To mitigate the costs associated with large antenna apertures and wide frequency bandwidths, the paper proposes the Radon Implicit Field Transform (RIFT), which integrates a classical forward model (Generalized Radon Transform) with an Implicit Neural Representation (INR) learned from radar signals. The method enables efficient scene representation and interpolation at unseen viewpoints, potentially reducing data collection burdens across various ASP applications. In addition, the paper introduces two novel error metrics to radar signal interpolation and evaluate performance. Experimental results demonstrate that RIFT achieves up to 188% improvement in scene reconstruction accuracy with only 10% of the data footprint compared to traditional models.

**Strengths:**

1. The paper learn implicit scene representations directly from radar signals, which is a new perspective for radar signal processing.
2. The proposed method can reconstruction and interpolate views using fewer measurements that non-deep learning methods.
3. The proposed two metrics are meaningful for the community.
4. The selected experiments show the efficiency of the proposed modules.

**Weaknesses:**

1. While the paper presents a novel method (RIFT) for radar signal processing, the experimental validation may not be comprehensive enough. Additional datasets or real-world scenarios could strengthen the claims regarding the method's effectiveness and robustness.
2.. The novelty of integrating deep learning methods with Generalized Radon Transform (GRT) is not well illustrated, what is the main contribution, only a combination of them?
3. The criteria and methodology for this benchmarking may require further elaboration. A more detailed description of how the benchmarks were selected and validated could enhance the paper's credibility.
4. The influence of hyperparameter selection on the performance of the INR model is not sufficiently explored.

**Questions:**

1. How do the data acquisition costs for radar imaging compare with those of other ASP applications, and what specific factors contribute to these differences?
2. How does the proposed Radon Implicit Field Transform (RIFT) manage to achieve scene reconstruction with only 10% of the data footprint compared to traditional models, and what specific mechanisms facilitate this reduction?
3. How does the joint benchmark for radar scene reconstruction and signal interpolation proposed in this study align with existing benchmarks in other imaging modalities?

---

> ### Author Response · Authors · 2024-11-23
> **Reply to Reviewer dWXL**
>
> Dear Reviewer dWXL,
>
> We sincerely appreciate your review and valuable feedback.
>
> Based on the weaknesses you pointed out, we made changes to the manuscript to improve the overall quality and credibility of the work.
>
> Here are the measures we took to address the weaknesses you mentioned:
>
> 1. Although we did not have access to a real-world radar dataset, we used commercial simulation software “Ansys Electronics Desktop 2023 R1” (referred to as “AEDT”), which is based on ray tracing and incorporates most of the effects encountered in real radar imaging tasks. Provided the reconstruction was not performed at very high resolution, our method outperformed the traditional least-squares-based radar imaging algorithm. Using scene rendering from the AEDT-generated data, we are confident in stating that we provided a first-order emulator of a scene with a significantly smaller data footprint.
>
> 2. We provided a code example (please refer to the general reply) and further elaboration in Appendix B.6 on the choice and structure of the metrics.
>
> 3. Indeed, hyperparameter tuning is challenging, especially when reconstructing scenes generated by AEDT. To address this, we added Appendix B.7 to discuss the thought process for tuning the hyperparameters.
>
> Here are the point-by-point answers to your questions:
>
> 1. Generating data from our forward models takes little time, as the GRT is a first-order estimation of the mapping from a scene to radar signals with no occlusion considered. Despite its simplicity, GRT is still widely used in radar imaging. For example, generating a scene with extents of 10m by 10m by 10m, a granularity of 0.2m, and 100 frequency points in the frequency band takes 7 minutes and 37 seconds on a graphics card, while commercial simulation takes 4 to 8 hours (depending on the ray tracing complexity). The difference in data generation speed primarily stems from the complexity of the forward model.
>
> 2. Typically, scene reconstruction from radar signals is achieved using ordinary least squares (OLS) methods. Without delving too deeply into least squares methods, the simplest explanation for RIFT’s superior performance is that the representation power of neural networks (NN) is significantly better than that of linear models. Additionally, least squares require considerable time due to the scale of the problem. The matrix we effectively invert is a 2D matrix with dimensions equal to (number of frequency samples × number of T/R pairs × number of viewpoints) by (number of voxels used to discretize the scene). This matrix can easily reach hundreds of GB in size, whereas the RIFT model checkpoint never exceeds 2MB. Thus, in addition to requiring less data to reconstruct the scene, RIFT also has smaller RAM requirements compared to the OLS baseline we used (though we did not experiment with other OLS solvers and cannot generalize this claim to all OLS solvers).
>
> 3. The analogy we would like to draw is as follows: m-SSIM and t-IoU are similar to metrics used in image segmentation. For example, if the segmentation is perfect, the inferred bounding box exactly aligns with the ground truth. m-SSIM measures how closely the reconstructed shape resembles the actual object, while t-IoU represents the IoU of the bounding boxes. Meanwhile, the p-RMSE of the radar signal is analogous to the loss function in NeRF (Mildenhall et al., 2020), which measures the error in the perceived signal (radar signal for RIFT and images for NeRF). The radar signal can be visualized as images of the scene using the signal processing techniques mentioned in this work (Na et al., 2006).
>
> We hope that our responses address your concerns and contribute to a potential improvement in your evaluation of our work.
>
> Best regards,
> Authors of Submission 12994
>
> References:
>
> 1. Ben Mildenhall, Pratul P. Srinivasan, Matthew Tancik, Jonathan T. Barron, Ravi Ramamoorthi, and Ren Ng. NeRF: Representing scenes as neural radiance fields for view synthesis. CoRR, abs/2003.08934, 2020. URL https://arxiv.org/abs/2003.08934.
> 2. Yibo Na, Yilong Lu, and Hongbo Sun, "A Comparison of Back-Projection and Range Migration Algorithms for Ultra-Wideband SAR Imaging," Fourth IEEE Workshop on Sensor Array and Multichannel Processing, 2006., Waltham, MA, USA, 2006, pp. 320-324, doi: 10.1109/SAM.2006.1706146.

---

> ### Comment · Reviewer_dWXL · 2024-11-26
>
> I am very grateful for your detailed reply and the efforts you have made in revising the manuscript. I have carefully considered your responses to the concerns I raised. It is evident that you have taken the review process seriously and have made reasonable attempts to address the identified weaknesses. After reviewing your responses, I find that they have enhanced my understanding of the work. Overall, my original assessment of the paper persists.

---

### Official Review · Reviewer_cpE5 · 2024-11-04

**Soundness:** 3
**Presentation:** 1
**Contribution:** 2
**Rating:** 5
**Confidence:** 3

**Summary:**

The paper presents the Radon Implicit Field Transform (RIFT), a novel method for reconstructing scenes from for synthetic aperture radar (SAR) imaging using an Implicit Neural Representation (INR). By encoding scene reflectivity in an INR, RIFT can reconstruct scenes with high fidelity from limited data samples, addressing the high data acquisition costs typical in SAR. The authors introduce custom error metrics—magnitude-SSIM (m-SSIM) for scene reconstruction and phase-RMSE (p-RMSE) for radar signal interpolation—to evaluate performance. Experimental results show that RIFT significantly improves scene reconstruction and viewpoint interpolation with reduced data compared to baseline models, demonstrating its potential for data-efficient radar-based imaging applications.

**Strengths:**

- The proposed method works well and outperforms existing approaches.
- Due to a lack of existing benchmarks, the authors created their own benchmark problems.  This can be a significant contribution if the authors make their benchmark open and easily accessible to the community.  I would encourage them to do so.

**Weaknesses:**

- Presentation of the paper could be improved.  Figures generally contains small plots which are sometimes difficult to inspect, some (e.g. Figure 5) are surrounded by a lot of white space, some are not discussed in the text.
- 3D scenes considered are very simple.

**Questions:**

- It is commented that a non-standard approach is taken of "accumulating gradients within an individual epoch across different views.... This gradient accumulation is specifically designed to mimic the physical motion inherent in synthetic aperture radar systems..."  This comment is somewhat cryptic and I did not fully understand why a non-standard approach was needed and, if so, how it mimics motion in SAR systems.  Could this comment please be elaborated and clarified.
- The abstract should stand alone and should not reference figures and tables in the paper.
- Figure 3 is not discussed in the text, as far as I could tell.
- Caption of Figure 4 is incomplete: "The data is"
- Typo p4, line 178: "scenarios, We"

---

> ### Author Response · Authors · 2024-11-23
> **Reply to Reviewer cpE5**
>
> Dear Reviewer cpE5,
>
> We sincerely appreciate your review and valuable feedback.
>
> Based on the weaknesses you pointed out, we have made changes to the manuscript to address the presentation issues you mentioned and to improve the overall quality of the work. In the supplemental materials, we have provided the code example used to generate the benchmarks, along with detailed comments.
>
> Here are the measures we took to address the weaknesses you mentioned:
>
> 1. Please refer to the overall comment for the changes made to improve the presentation of the paper (including the correction of typos). Additionally, in the supplemental materials, we included all vector graphics used in generating the figures. We hope these supplemental materials further enhance the readability of our work.
>
> 2. We used the commercial simulation software "Ansys Electronics Desktop 2023 R1" (referred to as “AEDT”), which is based on ray tracing and incorporates most of the effects encountered in real radar imaging tasks. In Section 4.3, we included a case study of reconstruction using AEDT data. Although the scene remains simplistic, the ray tracing forward generation can be considered as close to real radar data as possible, and we hope this supplemental section further strengthens the soundness of our work.
>
> Here are point-by-point answers to your questions (if applicable):
>
> 1. In the update, a paragraph was omitted because the statement does not apply to reproducing the scene generated by the AEDT proxy of a real-world scene. Please let us know if you are still interested in the details of accumulating gradient descent, and we would be happy to provide further explanation.
>
> 2. We relocated Figure 1. It was originally intended to serve as a teaser figure, but multiple reviewers expressed dissatisfaction with its placement.
>
> 3. We only mentioned the result of Figure 3 corresponding to Table 1. We added a brief discussion of it.
>
> 4. We corrected the typos you mentioned.
>
> We hope that our responses address your concerns and contribute to a potential improvement in your evaluation of our work.
>
> Best regards,
> Authors of Submission 12994

---

> > ### Comment · Reviewer_cpE5 · 2024-11-25
> >
> > I thank the authors for their thorough response and for making extensive revisions to the manuscript.  The additional AEDT case study is a welcome addition.  I have reviewed the comments by the other Reviewers and the Authors' responses.  On balance, my original assessment and rating of the paper stands.

---

> ### Author Response · Authors · 2024-11-27
> **Reply to Reviewer cpE5**
>
> Dear Reviewer cpE5,
>
> We appreciate your response.
>
> In a new comment to reviewer zohP, we have provided some details about
>     1. The implementation and the difference between radar imaging and optical imaging, and
>     2. Our justification on the reliability of the simulation tool we use.
> in the second reply to reviewer zohP. We believe this additional information can provide a different perspective on our work.
>
> We hope our comment can provide more useful information.
>
> Best,
> Authors of Submission 12994

---

### Author Response · Authors · 2024-11-23
**Record for the updated manuscript**

Dear Reviewers,

Thank you for your valuable time and suggestions to help us improve the quality of our work, Submission12994. Your feedback and questions are much appreciated. Below is a list of changes we made to the first draft to address some of your concerns.

Please note that the line numbers used to mark changes are from the initial submission dated Oct 12th. There might be slight differences in the line numbers in the new submission.

1. The most important change: We added Section 4.3, providing an example of the reconstruction of a simple scene from data generated by a commercial simulator based on ray tracing (Ansys Electronics Desktop 2023 R1, referred to as “AEDT,” which considers the multipath effect mentioned by reviewer zohP). This addition partially addresses the scene-reality correspondence problem that all reviewers raised. We are working on additional scenes generated by the same commercial simulator. We will investigate whether we are allowed to share the engineering files for the simulations and, if so, will make them available as well.

2. We added a link to the details about matrix $\mA$ in line 151 so that the description (in Section 3.1) and details (in Appendix B.3) of matrix $\mA$ appear where it is first mentioned. This addressed minor weaknesses noted by reviewer zohP and the last point of weakness 2 raised by reviewer B7gc.

3. We fixed a citation error where parentheses were missing in the previous version. The corrections were made in the citations on lines 136, 265, and 1017. This addressed minor weaknesses mentioned by reviewer zohP.

4. We fixed the typos mentioned by the reviewers, including typos on the following lines:

    179: Changed “2.2” to “Section 2.2.”
    210: Changed “Appendix B” to “Appendix B.4.”
    215: Added a period (“.”).

5. These changes addressed weaknesses noted by reviewers cpE5 and zohP.
6. We corrected all inconsistent unit expressions. Instead of using “x meters” and “xm” interchangeably, all units are now in the “xm” format. A footnote was added at the first instance. This partially addressed a weakness mentioned by reviewer zohP.

7. We rewrote the details of viewpoint sampling on lines 301 to 304 for clarification. This partially addressed weakness 2 noted by reviewer B7gc.

8. We deleted the paragraph on lines 220 to 226 because, in supplemental experiments, we found it is not always true. Without gradient accumulation across viewpoints, the reconstruction from data generated by AEDT was better.

9. We updated Section 5 to incorporate the discussion of a previous work mentioned by reviewer zohP. This addressed concerns about overclaims raised by reviewer B7gc and issues related to the discussion of related work noted by reviewer zohP.

10. We added equation numbers to all important equations, as suggested by reviewer B7gc.

11. We polished the writing throughout the manuscript.

Additionally, we will include a supplemental .zip file containing anonymized sample code for generating a scene, training a RIFT instance, and visualization. All the code is carefully documented. We also included vector plots for all figures provided in the manuscript.

In addition, we will provide another update by the end of the rebuttal period. Please kindly let us know if you have more concerns or suggestions that may further strengthen this work. We sincerely appreciate your reviews and critiques.

Best regards,
Authors of Submission12994


---
Upload Nov 24th:
1. We provide more details of the Ansys simulation tools in Appendix C.
2. We streamline the experimental design and the reference to different sections of Appendix B.

---

### Meta-Review · Area_Chair_hfcB · 2024-12-23

**Metareview:**

This paper proposes RIFT for SAR (Synthetic Aperture Radar), which is considered a special case of ASP (Array Signal Processing). Data acquisition for SAR, and ASP more generally, is notoriously challenging, and the authors posit that INR (Implicit Neural Representation) methods could help address these issues. Using synthetic data to train an INR-based model, the authors report improved performance across various metrics, including newly proposed ones. Reviewers offered multiple suggestions for improvement, prompting the authors to revise the manuscript. However, the key concern shared by the reviewers involves the quality and scope of the synthetic data, as well as the gap between synthetic and real data. Although the authors relied on commercial tools for data synthesis, they could not fully resolve the reviewers’ concerns about this gap. The AC agrees that the overall pipeline is relatively straightforward and that demonstrating strong generalization to real-world scenarios is critical for acceptance. Therefore, the AC has decided to reject this manuscript in its current form.

**Additional Comments On Reviewer Discussion:**

A key concern raised by reviewers is the lack of verification with real-world data. It appears that the authors have made significant efforts, such as using professional simulation tools. AC agrees that not all papers require verification with real data. Nevertheless, the generalization capabilities of INR-based methods are crucial; without such validation, it is difficult to assess the practical effectiveness of the proposed method.

---

### Decision · Program_Chairs · 2025-01-22

Reject